# Temozolomide-Acquired Resistance Is Associated with Modulation of the Integrin Repertoire in Glioblastoma, Impact of α5β1 Integrin

**DOI:** 10.3390/cancers14020369

**Published:** 2022-01-12

**Authors:** Saidu Sani, Nikita Pallaoro, Mélissa Messe, Chloé Bernhard, Nelly Etienne-Selloum, Horst Kessler, Luciana Marinelli, Natacha Entz-Werle, Sophie Foppolo, Sophie Martin, Damien Reita, Monique Dontenwill

**Affiliations:** 1Laboratoire de Bioimagerie et Pathologies (LBP), UMR CNRS 7021, Faculté de Pharmacie, Université de Strasbourg, 67401 Illkirch, France; Sanisaidu96@gmail.com (S.S.); npallaoro@unistra.fr (N.P.); melissa.messe@etu.unistra.fr (M.M.); chloe.bernhard@etu.unistra.fr (C.B.); nelly.etienne-selloum@unistra.fr (N.E.-S.); Natacha.Entz-Werle@chru-strasbourg.fr (N.E.-W.); sophie.foppolo@unistra.fr (S.F.); sophie.martin@unistra.fr (S.M.); damien.reita@gmail.com (D.R.); 2Cancer and Diabetic Research Group, Department of Biochemistry, Faculty of Biological Science, Federal University Ndufu-Alike Ikwo, Abakaliki 482131, Nigeria; 3Department of Pharmacy, Institut de Cancérologie Strasbourg Europe (ICANS), 67200 Strasbourg, France; 4Department of Chemistry, Institute for Advanced Study, Technical University of Munich, 85748 Garching, Germany; kessler@tum.de; 5Dipartimento di Farmacia, Università degli Studi di Napoli “Federico II”, 80131 Naples, Italy; lmarinel@unina.it; 6Pediatric Onco-Hematology Unit, University Hospital of Strasbourg, 67098 Strasbourg, France; 7Département de Génétique Moléculaire des Cancers, Oncologie Moléculaire & Pharmacogénétique, University Hospital of Strasbourg, 67000 Strasbourg, France

**Keywords:** glioblastoma, temozolomide resistance, integrins, p53 reactivation

## Abstract

**Simple Summary:**

Glioblastomas are the deadliest brain tumours. The standard of care associates surgery, radio- and chemotherapy with Temozolomide as the reference drug. Despite this treatment, most of the tumours recur. The characterization of resistance mechanisms is of paramount importance to enable the proposal of more effective therapies. In this work we aimed to evaluate the molecular changes occurring during and after Temozolomide treatment in a glioma cell line. A high plasticity in the integrin repertoire exists in these cells. As an example, variations of the α5β1 integrin expression were observed with a reduction during the treatment and re-expression after removal of the drug. The association of integrin antagonists with p53 reactivators appears to be efficient in recurrent tumours. Specific integrins may thus be particularly targetable at different time points of glioblastoma treatment and combination therapies evaluated according to their time-dependent expression.

**Abstract:**

Despite extensive treatment, glioblastoma inevitably recurs, leading to an overall survival of around 16 months. Understanding why and how tumours resist to radio/chemotherapies is crucial to overcome this unmet oncological challenge. Primary and acquired resistance to Temozolomide (TMZ), the standard-of-care chemotherapeutic drug, have been the subjects of several studies. This work aimed to evaluate molecular and phenotypic changes occurring during and after TMZ treatment in a glioblastoma cell model, the U87MG. These initially TMZ-sensitive cells acquire long-lasting resistance even after removal of the drug. Transcriptomic analysis revealed that profound changes occurred between parental and resistant cells, particularly at the level of the integrin repertoire. Focusing on α5β1 integrin, which we proposed earlier as a glioblastoma therapeutic target, we demonstrated that its expression was decreased in the presence of TMZ but restored after removal of the drug. In this glioblastoma model of recurrence, α5β1 integrin plays an important role in the proliferation and migration of tumoral cells. We also demonstrated that reactivating p53 by MDM2 inhibitors concomitantly with the inhibition of this integrin in recurrent cells may overcome the TMZ resistance. Our results may explain some integrin-based targeted therapy failure as integrin expressions are highly switchable during the time of treatment. We also propose an alternative way to alter the viability of recurrent glioblastoma cells expressing a high level of α5β1 integrin.

## 1. Introduction

Glioblastoma (GBM) is the most common and most aggressive malignant brain tumour in adults and is characterized by high proliferation, invasion into normal brain tissue and resistance to therapies [1,2]. Currently there is no effective long-term treatment for this killer disease, but the standard of care (Stupp protocol) is maximal surgical resection, followed by radiotherapy with concomitant and adjuvant Temozolomide (TMZ) chemotherapy [3]. However, the prognosis of patients with glioblastoma remains poor and has not improved despite numerous clinical trials on new therapeutic propositions [4]. Therefore, there is an urgent need for novel therapeutic strategies, which may be supported by a deep understanding of GBM and surrounding microenvironment crosstalk.

Primary and acquired resistances are the major challenges for the clinical use of standard and targeted therapies in GBM [5,6]. One of the mechanisms of glioblastoma resistance to TMZ involves O^6^-methylguanine DNA methyl transferase (MGMT), a suicide enzyme that allows the direct repair of the lesions caused by TMZ, through the removal of a methyl group in position O^6^ of guanine [7]. Previous studies have shown that intracellular accumulation of the tumour suppressor protein p53 downregulates the expression of MGMT [8,9]. The p53 protein is mutated in about 30% of glioblastomas [10], although a subtype of glioblastoma expressing mostly wild type p53 has been identified in an integrated genomic analysis [11]. However, even if wild type p53 is expressed in 70% of GBM, its functions are frequently suppressed by MDM2/MDM4, E3 ubiquitin ligases that mark and target p53 for proteasomal degradation [12,13,14]. MDM2, a zinc finger nuclear phosphoprotein and negative regulator of the p53 protein, is often overexpressed in GBM and has been implicated in cancer cell proliferation and survival. Inhibition of the p53–MDM2 interaction can prevent p53 degradation and restore p53 transcriptional activity, leading to the p53-mediated induction of tumour cell apoptosis, thus making the p53–MDM2 complex a promising target for glioblastoma expressing wild type p53 [15,16,17]. The discovery of the p53–MDM2 inhibitor Nutlin-3a represented a breakthrough in the development of p53-activators ^14^. The more recent development of RG7388 (Idasanutlin), a second-generation MDM2 inhibitor with greater potency, selectivity, bioavailability and effective p53-activating ability leading to the p53-mediated induction of tumour cell apoptosis, is promising for cancers, including GBM [18,19].

Integrins have become, in the past 20 years, the subject of numerous studies because of the vital role they play in tumour progression [20,21]. The integrins are transmembrane heterodimeric cell surface receptors that mediate cell adhesion to the extracellular matrix (ECM) and support cell–cell interactions in multiple physiological and pathological conditions [22,23]. The frequent deregulation of integrin expressions and pathways in cancer cells underscores specific integrin major contributions in tumour growth and resistance to therapies [24]. The disruption of the integrin signalling pathways by integrin antagonists has been shown to inhibit tumour growth and sensitize tumours to therapies in preclinical contexts. Cilengitide [25] was the first integrin αvβ3/β5 antagonist to reach the clinic, but clinical trials for GBM with cilengitide in combination with the standard of care (Stupp protocol) were unsuccessful [26]. Several reasons may explain these failures, as reviewed in [27]. However, knowledge on integrin expressions and functions in GBM merits further investigations to adapt integrin-based therapies to specific subpopulations of patients. In this way, expression of α5 subunit of the α5β1 integrin heterodimer is enhanced in the mesenchymal subgroup of glioblastoma patients as compared to the others [11,28], presumably conferring to these patients a better sensitivity to anti-α5 integrin therapy. We and others showed previously that α5β1 integrin is an interesting therapeutic target for GBM. Its overexpression at the mRNA [29,30] or protein [31] levels define populations of patients with worse prognostics. In preclinical experiments it was shown that this integrin is involved in survival, migration, resistance to therapies and neo-angiogenesis, all being hallmarks of GBM [32,33,34,35].

Integrin expressions in GBM vary after therapies [36]. As an example, it was shown that in tumours recurring after bevacizumab treatment, α5β1 integrin was overexpressed in a subpopulation of patients [37]. In particular, we demonstrated the existence of a negative crosstalk between α5β1 integrin and p53 pathways supporting an implication in glioma resistance to chemotherapies [30,38]. We also showed that the inhibition of the integrin concomitantly with p53 activation with Nutlin 3a in α5-overexpressing cells led to a huge increase in cell apoptosis [39].

In this study, we aimed to investigate if Temozolomide treatment affects the integrin repertory in glioma cells, taking the U87MG cell line as an example. Resistant cells were obtained that conserve the resistance even after removal of the drug. Transcriptomic analysis of non-treated U87MG cells and resistant cells cultured in the presence or absence of TMZ showed a high variation in integrin expressions. Interestingly, α5β1 integrin expression decreases in the presence of the drug but recovers after its removal, suggesting that it may represent a therapeutic target for recurrent glioblastoma. We also investigated if treatment of these recurrent TMZ-resistant cells may be sensitive to a combination of highly active and selective α5β1 integrin antagonists and p53 reactivators. Our results may add new therapeutic perspectives for recurrent glioblastoma expressing high level of α5β1 integrin.

## 2. Materials and Methods

### 2.1. Drugs

Temozolomide (TMZ), 8-carbamoyl-3-methylimidazo[5,1-d]-1,2,3,5-tetrazin (3H)-one 5 (Sigma-Aldrich), was prepared as a 100 mM stock solution in DMSO and stored at 4 °C until use. Nutlin-3a, (4-[4,5-bis-(4-chlorophenyl)-2-(2-isopropoxy-4-methoxy-phenyl)-4,5-dihydro-imidazole-1-carbonyl]-piperazin-2-one), the active enantiomer, was from Cayman chemical company (Interchim, Montluçon, France). Nutlin-3a was prepared as a 10 mM stock solution in ethanol and stored at −20 °C until use. Idasanutlin, also known as RG7388 (C_31_H_29_C_l2_F_2_N_3_O_4_), was from Euromedex (Souffelweyersheim, France). Idasanutlin was prepared as a 10 mM stock solution in DMSO and stored at −20 °C until use. RITA (5,5′- (2,5-furandiyl) bis-2 thiophenemethanol) was from Cayman chemical company (Interchim, France). RITA was prepared as a 10 mM stock solution in ethanol and stored at −20 °C until use. Antagonists of α5β1 integrin, 34c and 1b (respectively named K34C and FR248 in this work) were synthesized according to the procedure described in [40,41]. Their structures and binding activities for integrin α5β1 and integrin αvβ3 are shown in [34]. Compound **9** was prepared as described in [42]. They were prepared as 10 mM stock solutions in DMSO and stored at 4 °C until use.

### 2.2. Cell Culture

U87MG glioblastoma cell line (p53 wild type) was from American Type Culture Collection (LGC Standards Sarl, Molsheim, France). Cell lines were routinely cultured in EMEM (Minimum Essential Medium Eagle) supplemented with 10% heat-inactivated foetal bovine serum, 1% Na pyruvate and 1% non-essential amino acids at 37 °C in a humidified atmosphere containing 5% CO_2_. All cells were periodically authenticated by Multiplexion GmbH and tested for the presence of mycoplasma.

### 2.3. Generation of Temozolomide-Resistant Glioblastoma Cells

U87MG cells (500,000) were seeded in T25 cm^2^ cell culture flask containing EMEM (Eagle’s minimal essential medium) supplemented with 10% serum, 1% Na pyruvate and 1% non-essential amino acids. Cells were allowed to adhere for 24 h, and the medium was replaced with a fresh medium containing 50 µM of TMZ. Cell treatment was repeated twice a week for several weeks, resulting in a sub-population of stable TMZ-resistant cells. The sub-population of resistant cells generated were continuously cultured in a medium containing TMZ and named U87MG R50. After two months, a subpopulation of U87MG R50 cells was cultured in medium without TMZ and named U87MG R50 OFF (or U87MG OFF). The maintenance of resistance in this cell line is checked regularly (every three months) by the Incucyte technology and resistance has appeared stable for at least one year.

### 2.4. IncuCyte Cell Confluence Assay as an Index of Proliferation

Cells were plated (2000 cells in 100 μL per well) into 96-well culture plates. The control solvent or drugs at 2× concentration in 100 μL of 2% FBS-containing medium was added to the appropriate wells. To monitor cell growth, the plates were placed into the IncuCyte live-cell analysis system and allowed to warm at 37 °C for 30 min prior to standard scanning with 4× objective every 3 h for at least 96 h. The captured phase contrast images were analysed using the IncuCyte ZOOM software provided by the manufacturer.

### 2.5. Senescence Assay

The β-galactosidase activity at pH 6 was determined using the Senescence Cells Histochemical Staining Kit (Sigma-Aldrich, Saint-Quentin-Fallavier, France) according to the manufacturer’s instructions. Briefly, cells were plated (2000 cells/200 µL) into a 96 well plate. The cells were washed twice with 1xPBS and fixed with 1× fixation buffer for 7 min at room temperature. The fixation buffer was aspirated and wells rinsed thrice with 1 × PBS. After washing, the cells were covered with staining mixture and incubated at 37 °C without CO_2_. After 12 h of staining, light microscopy was used to identify senescent (blue-stained) cells.

### 2.6. Western Blotting

Cells were plated (200,000 cells per well) into 6-well culture plates and treated with the control solvent or drugs. For the basal level values of proteins of interest, cells were used 24 h after plating. Proteins were extracted from adherent and floating cells after lysis with Laemmli sample buffer (Bio-Rad, Marnes La Coquette, France) on ice and lysates heated at 95 °C for 10 min. Proteins were separated on precast gradient 4–20% SDSPAGE gels (Bio-Rad) and transferred to PVDF membrane (GE Healthcare, Velizy, France). After 1 h of blocking at room temperature, membranes were probed with appropriate primary antibodies (Table 1) overnight at 4 °C. Membranes were subsequently incubated with anti-rabbit or anti-mouse antibodies conjugated to horseradish peroxidase (Promega, Charbonnieres les- Bains, France), developed using chemoluminescence (ECL, Bio-Rad) and visualized with Las4000 image analyser (GE Healthcare, France). Quantification of non-saturated images was done using ImageJ software (National Institutes of Health, Bethesda, MD, USA). GAPDH or tubulin was used as the loading control for all samples.

### 2.7. RNAseq Data

RNA-Seq libraries were generated from 400 ng of total RNA using TruSeq Stranded mRNA Library Prep Kit and TruSeq RNA Single Indexes kits A and B (Illumina, San Diego, CA, USA), according to manufacturer’s instructions. Briefly, following purification with poly-T oligo-attached magnetic beads, the mRNA was fragmented using divalent cations at 94 °C for 2 min. The cleaved RNA fragments were copied into first-strand cDNA using reverse transcriptase and random primers. Strand specificity was achieved by replacing dTTP with dUTP during second-strand cDNA synthesis using DNA Polymerase I and RNase H. Following addition of a single ‘A’ base and subsequent ligation of the adapter on double-stranded cDNA fragments, the products were purified and enriched with PCR (30 s at 98 °C; (10 s at 98 °C, 30 s at 60 °C, 30 s at 72 °C) × 12 cycles; 5 min at 72 °C) to create the cDNA library. Surplus PCR primers were further removed by purification using AMPure XP beads (Beckman Coulter, Villepinte, France), and the final cDNA libraries were checked for quality and quantified using capillary electrophoresis. Libraries were then sequenced on an Illumina HiSeq4000 system as single-end 1 × 50 base reads. Image analysis and base calling were performed using RTA 2.7.7 and bcl2fastq 2.17.1.14. Reads ((Illumina, San Diego, CA, USA)) were preprocessed using Cutadapt version 1.10 in order to remove adapter, polyA and low-quality sequences (Phred quality score below 20), and reads shorter than 40 bases were discarded for further analysis. Reads mapping to rRNA were also discarded (this mapping was performed using Bowtie version 2.2.8). Reads were then mapped onto the hg38 assembly of human genome using STAR version 2.5.3a (twopassMode Basic). Gene expression was quantified using htseq-count version 0.6.1p1 and gene annotations from Ensembl release 99. Statistical analysis was performed using R 3.3.2 and DESeq2 1.16.1 Bioconductor library. Sequencing was performed by the GenomEast platform, a member of the ‘France Génomique’ consortium (ANR-10-INBS-0009).

Gene expression data obtained with DESeq2 were used to generate the heatmaps and dendrogram with R. Only genes expressed in all conditions (defined if normalized reads count divided by median of transcripts length in kb is greater than 1 for gene across all nine libraries) were taken into account for visualization. Hierarchical clustering method was performed according to pairwise complete-linkage method and using Pearson correlation for row clustering and Spearman correlation for column clustering. The biological significance of differentially expressed genes (DEGs) obtained was explored using ReactomePA, an R/Bioconductor package for reactome pathway analysis and visualization. |Log2 (fold change) | > 2 (|log2FC|>2) and an adjusted false discovery rate (FDR) < 0.05 (using Benjamini–Hochberg correction) were used as the cut-off criteria of DEGs samples.

### 2.8. Confocal Microscopy and Image Analysis

Coverslips were coated with fibronectin (20 μg/mL in DPBS), and 25,000 cells were seeded in 10% serum containing medium and cultured for 24 h. Cells were then fixed in 4% (*v*/*v*) paraformaldehyde for 12 min and permeabilized with 0.1% Triton-X100 for 2 min. After 1 h of blocking step in 3% bovine serum albumin (BSA)-PBS solution, cells were incubated with primary antibodies against integrin α5 (clone IIA1 Pharmingen, 1 μL/100 μL in 3% PBS-BSA) and β1 integrin (purified anti-human CD29, Clone:TS2/16, 1 μL/100 μL in 3% PBS-BSA) overnight at 4 °C. Cells were rinsed in 1 × PBS and incubated with appropriate secondary antibodies (Alexa fluor@ 647 goat anti-mouse (A21236) 1 μL/200 μL in 3% PBS-BSA and Alexa fluor@ 688 goat anti-mouse 1 μL/200 μL in 3% PBS-BSA) and 4′, 6-diamidino-2-phenylindole (DAPI) (1 μL/2000 μL in 3% PBS-BSA was added for nuclei staining) for 45 min. Samples were mounted on microscope slides using fluorescence mounting medium (Dako). Images were acquired using a confocal microscope (LEICA TCS SPE II, 63× magnification). For each experiment, identical background subtraction and scaling was applied to all images. Pearson correlation coefficient from 10–12 images (4–5 cells per images) from 3 independent experiments was calculated using JACoP plugin ImageJ software.

### 2.9. Spheroid Migration Assays

A single-cell suspension was mixed in MEM supplemented with 10% foetal bovine serum, 1% Na pyruvate, 1% non-essential amino acid containing 20% of methylcellulose. Spheroids were made by the hanging drop method with 2000 cells in a 20 μL drop as previously described [43]. Tissue culture plates were coated with fibronectin or polylysine (10 μg/mL in DPBS solution) for 3 h at 37 °C. Two-day-old spheroids were allowed to adhere to fibronectin-coated plates and migrate in complete medium (MEM, 10% FBS, 1% Na pyruvate and 1% non-essential amino acids) either with solvent or supplemented with 20 or 5 µM of K34C or FR248. Eighteen hours later, cells were fixed with glutaraldehyde 1% (Electron Microscopy Sciences) and stained with DAPI diluted at 1 µg/mL in 3% PBS-BSA (Sigma-Aldrich). Nuclei were picturized under the objective 5× in the fluorescence microscope ZEISS-Axio (ZEISS). To evaluate the number of cells that migrated out of the spheroid and the average distance of migration out of the spheroid, image analysis was performed with ImageJ software using a homemade plugin [43]. Phase-contrast images (EVOS Xl, Core10× magnification, Thermo Scientific) were also acquired.

### 2.10. Statistical Analysis

Statistical analyses were performed using Student’s *t*-test with GraphPad Prism software. All data are presented as mean ± SEM from three or more independent experiments with *, *p* < 0.05; **, *p* < 0.01; ***, *p* < 0.001.

## 3. Results

### 3.1. Long-Term Exposure of U87MG Cells to TMZ Generates Persistent Resistant Cells

To generate TMZ-resistant cells, we subjected U87MG cells to 50 µM of TMZ for several weeks, resulting in a sub-population of stable TMZ-resistant U87MG cells. The sub-population of resistant cells was either continuously cultured in the medium containing 50 µM of TMZ (U87MG R50) or, after 2 months in the presence of TMZ, cultured in the medium without TMZ (U87MG R50 OFF). We confirmed that the parental cells were sensitive to 50 μM of TMZ and that the U87MG R50 cells were insensitive to TMZ for up to 96 h (Figure 1A). Interestingly, TMZ resistance was maintained in U87MG R50 OFF cells (Figure 1A). Furthermore, while the parental cells were dose-dependently sensitive to TMZ, the U87MG R50 and U87MG R50 OFF cells remained insensitive to varying concentrations up to 100 μM of TMZ (Appendix A).

Phase-contrast images obtained from the IncuCyte (Figure 1B) showed that the cellular morphology of U87MG R50 cells differed from their parental cells by having a more spread, enlarged and flattened morphology. The U87MG R50 OFF cells displayed a mixed morphology, with some cells presenting the morphology of U87MG R50 cells while the others presented the morphology of the parental cells. TMZ treatment for 3 days produced a profound change in the cellular morphology of the parental cells, causing extensive branching, reduction in total cell number and confluence (Figure 1B). On the contrary, both the U87MG R50 cells and U87MG R50 OFF cells appeared healthy, with no obvious morphological changes 72 h post TMZ treatment (Figure 1B). Overall, these results demonstrate that a fraction of the U87MG cells that survived prolonged exposure to TMZ developed acquired TMZ-resistance, which was maintained even after the withdrawal of TMZ. According to the sample RNAseq data (see below), this resistance is not due to MGMT expression as no increase in MGMT mRNA was observed.

### 3.2. TMZ Induces Senescence in U87MG Cells

Since the resistant cells showed the morphological features of senescent cells, we next tested the senescence marker SA-βGAL. Upon short incubation in 50 μM of TMZ, the remaining cells were all positive for SA-βGAL staining. Positive cells were maintained in the U87MG R50 cells but largely decreased in U87MG R50 OFF cells (Figure 2A). Accordingly, a significant increase in the expression of proteins p16 and p21, known to be involved in senescence, was observed in U87MG R50 cells as compared to the parental cells. An increase in p16 but not in p21 was recorded in U87MG R50 OFF cells (Figure 2B). Altogether, the results confirm that TMZ triggers senescence in U87MG cells, as already reported in [44], which is resumed after TMZ removal.

### 3.3. Extracellular Matrix Organization and Integrins Are Affected by TMZ

We used RNAseq analysis to compare the three cell lines. Analysis of global gene variations by unsupervized hierarchal clustering showed that profound changes occurred during TMZ treatment, with the largest differences recorded between U87MG and U87MG R50 cells (Figure 3A). U7MG R50 OFF were more closely related to the U87MG non-treated cells. The most significant genes were subjected to gene ontology analysis. Biological pathways mostly impacted by long-term Temozolomide treatment (R50 versus control cells) are extracellular matrix organization, O-linked glycosylation and integrin cell surface interactions (Figure 3B). By contrast fewer biological pathways, including neuronal system and extracellular Matrix organization, appear to be impacted in U87MG R50 OFF cells compared to non-treated cells (Figure 3C).

Focusing on the integrin genes, profound rearrangements of integrin subunit expression levels were observed (Figure 4A). Four main clusters exist, which define particular integrin expressions in the different cell lines. Cluster 1 corresponds to integrins expressed in U87MG control cells but repressed in R50 cells and with intermediary expression in R50 OFF cells (Integrins α2, α3, α4, α5, α10, α11, αv, β1, β3). Cluster 2 involves integrins overexpressed in U87MG R50 OFF cells as compared to the two other cell lines (α6, β6 and β8), of which α6 and β8 are described as markers of glioma stem cells [36,45,46]. Cluster 3 corresponds to integrins repressed in U87MG R50 OFF cells (α7, α9, αx, αD, β5, β7). Cluster 4 includes integrins overexpressed in U87MG R50 cells (α1, αL, αM, β2, β4), including some leucocyte specific integrins. Data confirm that integrin mRNA expressions are subjected to specific variations during the time course of chemotherapy with Temozolomide.

As we demonstrated previously that integrin α5β1 is involved in TMZ resistance [30], we next focused on this integrin and examined its expression at the protein level in the parent and TMZ-resistant cells lines. By Western blot, we observed a clear decrease in the α5 subunit in U87MG R50 cells without a significant change in that of the β1 subunit. Interestingly, the α5 integrin level increased after removal of TMZ in the U87MG R50 OFF cells, reaching those seen in non-treated cells without changes in the β1 integrin level (Figure 4B). By contrast, the αv integrin level did not change between the three cell lines, although β3 integrin expression followed those of α5 integrin, showing that both α5β1 and αvβ3 integrins, which are largely involved in glioblastoma aggressiveness, were affected by Temozolomide treatment. As a confirmation we checked the expression changes of α5β1 integrin by immunohistochemistry. As can be seen in Figure 4C, the expression of both α5 and β1 integrins was reduced in U87MG R50 but reappeared after removal of TMZ in U87MG R50 OFF cells.

### 3.4. TMZ Affects Glioma Cell Proliferation and Migration through Modulation of α5β1 Integrin

The capacity to fill the culture wells as well as the proliferation index was slightly decreased for U87MG R50 cells as compared to parental and U87MG R50 OFF cells (Figure 5A) presumably due, at least in part, to the decrease in integrins. To confirm the impact of α5β1 integrin on proliferation, we used the integrin antagonists K34c and FR248, which are RGD-based peptidomimetics optimized for high affinity to α5β1 integrin with a reduced affinity to αvβ3 integrin. We already showed that both antagonists recognize α5β1 integrin and inhibit glioma cell adhesion to fibronectin and cell migration [35]. As shown in Figure 5B, K34c proved able to decrease the cell adherence for all three cell lines, inhibiting the cell spreading and forming some sphere-like structures, particularly in U87MG and U87MG R50 OFF cells. FR248 is less efficient in inhibiting cell spreading than K34c and particularly ineffective in U87MG R50 cells. A quantification of cell confluence after 3 days of treatments confirmed these morphological observations (Figure 5C). FR248, which is slightly more selective than K34c for α5β1, is inactive in U87MG R50 cells in accordance with the low level of α5β1 integrin in these cells. The data suggest that U87MG and U87MG R50 OFF cells may be similarly sensitive to α5β1 integrin antagonists at least for the inhibition of cell adherence and confluence.

Integrins are largely involved in glioma cell migration, as we have shown previously for α5β1 integrin [33,34,43]. The capability of cell dispersion out of gliomaspheres in a fibronectin-rich environment was then compared among the three cell lines. As shown in Figure 6, U87MG and U87MG R50 OFF cells were able to disseminate out of the spheres, but dissemination was blocked in U87MG R50 cells. This dissemination was clearly impacted by the expression of α5β1 integrin as it was largely inhibited by K34c and FR248 at 20 µM (Figure 6) but also at 5 µM (Appendix A) in U87MG and U87MG R50 OFF cells. TMZ acute treatment of U87MG and U87MG R50 OFF cells did not affect cell migration, nor did TMZ removal affect U87MG R50 cells (Appendix A). Both cell lines were unable to migrate strongly on polylysine, a nonspecific substrate, as compared to fibronectin, the privileged ECM substrate of α5β1 integrin (Appendix A). The results show that TMZ-resistant U87MG R50 OFF cells recover their capacity not only to proliferate but also to migrate on a fibronectin-rich substrate.

### 3.5. p53 Signalling Pathway in U87MG and TMZ-Resistant Cells

TMZ is known to activate the p53 pathway. We checked p53 stabilisation and activation in the three cell lines in the basal conditions of the culture. In U87MG R50 cells as well as in the U87MG R50 OFF cells, p53 is stabilized, and its target genes MDM2 and p21 are both increased at the mRNA level in the former cell line but only MDM2 in the latter one (Figure 7A). These data are confirmed by the protein level of both p21 (see Figure 2B) and MDM2 (Figure 7A). MDM2 may thus be used as a target in these resistant cells. We considered three inhibitors of MDM2/p53 interactions already known to reactivate p53 signalling. The different molecules used were Nutlin-3a and Idasanutlin, which bind to the MDM2 part and RITA, which binds to the p53 part of the MDM2–p53 complex. As can be seen in Figure 7B, all three drugs enhanced the stability of p53 as well as its activation (shown by the increase in MDM2 protein). No differences could be observed between control U87MG cells and their TMZ-resistant counterparts as far as the activation of p53 is concerned. In these experiments, Idasanutlin was the most efficient activator of the p53 pathway, even at 0.1 µM (a concentration 10 times lower than for Nutlin-3a and RITA), and RITA was less efficient after 24 h of treatment. The results suggest that TMZ-resistant cells may benefit from an alternative way to activate the p53 tumour suppressor pathway.

### 3.6. p53 Activation and Integrin Inhibition as a Therapeutic Option for TMZ-Resistant Cells

We described previously negative crosstalk between α5β1 integrin and p53 signalling pathways implicated in the chemotherapy resistance of glioma cells. We showed that activating p53 concomitantly with inhibiting integrin α5β1 led to an increase in p53 signalling in α5 integrin subunit overexpressing cells [39]. We thus wondered if similar results might be obtained in TMZ-resistant cells. We investigated the association of p53 activators Nutlin-3a and Idasanutlin with the two integrin antagonists on cell confluence. Data are summarized in Table 2. As an example, results after 72h of treatment with Idasanutlin (0.1 µM) in association with K34c (20 µM) or FR248 (20 µM) are shown in Figure 8A. In addition, the p53 signalling pathway appears over-activated in U87MG R50 OFF cells after the association of Idasanutlin with FR248 (Figure 8B).

Compound **9** was described as a potent α5β1 and αvβ3 integrin inhibitor coupled with an inhibitory activity on MDM2 and MDM4, thus combining the effects we were studying [42]. We thus checked the effect of compound **9** in our cell lines. It was highly effective in inhibiting cell confluence in both U87MG and U87MG R50 OFF, cells even at low doses with 50% inhibition around 0.6 µM and profound changes on cell morphology, as shown in Figure 9A,B. Low doses of compound **9** (0.1 to 0.6 µM) hardly affected U87MG cells that were knocked down for the α5 integrin gene by the CRISPR/Cas9 technology (U87MG α5KO cells), confirming its capability to recognize this integrin (Appendix A). Interestingly, in low doses, compound **9** (0.6 µM) behaved as integrin inhibitor K34c treatment alone but in high doses (above 5 µM), it had similar effects to the association of K34c with p53 activators in U87MG and U87MG R50 OFF cells (compare with Figure 8A). It seems that high doses are needed to activate p53 target genes (data not shown). The results suggest that compound **9** behaves as a strong inhibitor of naïve and TMZ-resistant U87MG cells through the concomitant inhibition of integrins and MDM2/MDM4.

## 4. Discussion

Although TMZ is currently the only approved chemotherapeutic drug known to significantly improve the overall survival of GBM patients [47,48], the development of acquired TMZ resistance leading to treatment failure remains one of the challenges to be resolved. Numerous works aimed to understand intrinsic and acquired TMZ resistance and recent reviews dedicated to this topic are available [49,50]. It appears clear that a multifaceted view is to be considered related to the high molecular heterogeneity of GBM and the plasticity of GBM cells. A consensus already exists about the role of MGMT, through which epigenetic regulation (promoter methylation or demethylation) is involved in the clinical response to TMZ. However, GBM-expressing or non-expressing MGMT can develop resistance to TMZ. Acquired resistance was also often linked to DNA damage repair pathways leading to new therapeutic avenues [51].

In this work, we generated TMZ-resistant cells by subjecting U87MG cells to TMZ 50 µM treatment as an in vitro model of MGMT-negative TMZ resistance. We aimed to compare cells continuously grown in the presence of TMZ (U87MG R50) with resistant cells growing in the absence of TMZ (U87MG R50 OFF), using this last model as a reflection of clinical recurrence. We confirmed that U87MG cells were sensitive to TMZ, resulting in large percentage of cell death at the treatment, beginning with few cells remaining alive but exhibiting hallmarks of senescence. Senescence is considered a favourable response to chemotherapy as it blocks tumoral cell proliferation. This view has been challenged as the irreversibility of drug-induced senescence remains under debate and the pro-tumorigenic properties of the senescent-associated secretory phenotype (SASP) are clearly demonstrated [52,53]. It has already been demonstrated that TMZ induces senescence rather than apoptosis in glioma cells [44,54,55]. In line with this, we demonstrated that the few U87MG cells that remained alive after one-week incubation with TMZ were all positive for SA-βGAL staining. In our experimental conditions, the regrowth of the few surviving cell population occurred in the presence of TMZ from these stained cells, suggesting that senescence was reversible, but this point remains to be confirmed. The regrowth of U87MG cells after TMZ treatment to obtain resistant cell populations is largely described in the literature. However, treatments followed different experimental procedures (with large variations in the chosen doses, duration of treatments, time of omics or phenotypic evaluations, etc…) that may result in different biological outcomes [50]. The characterization of TMZ-resistant cells was generally made at the endpoint of cell treatment rather than along the treatment protocol. Recently the development of resistance was studied in a glioma cell line. Interestingly, a transient state (from day 3 to 9 after treatment) defined by slow growth and morphological and metabolic changes was characterized. Resistant cells will emerge from these transient state cells [56]. The link with senescence has not been studied in this work.

In our work we aimed to analyse another step of the TMZ resistance, i.e., how resistant cells behave after removal of the drug, to gain an understand of what may happen in patients before or at the point of recurrence. To he best of our knowledge, the molecular and phenotypic characterization of resistant cells before and after removal of the drug has not been extensively studied. Interestingly, profound changes were observed between U87MG R50 cells and the non–treated cell, whereas U87MG R50 OFF cells showed more closely related characteristics to the control cells. One of the most affected pathways was, in both cases, extracellular matrix organization, including integrin expression level modifications. Interestingly, profound changes in ECM–receptor interactions were also noted in the response of the glioma to ionizing radiation [57]. The heatmap of integrins (Figure 4) revealed particular sets of integrins overexpressed in each cell line, suggesting that anti-integrin therapeutic options have to be considered in a timely manner during therapies. As an example, U87MG R50 OFF cells overexpress α6 and β8 integrins, both known to be glioma stem cell markers [36,45]. Accordingly, a dedifferentiation of differentiated cells towards glioma stem cells has been reported in tumours after radiotherapy or TMZ chemotherapy [58,59]. Specific anti-α6 and/or β8 integrin therapies may thus be used for recurrent as well as for primary tumours.

In this work we focused on RGD-integrins such as α5β1 and αvβ3, as we demonstrated in previous studies that α5β1 integrin is involved in TMZ resistance [30]. We found that continuous treatment with TMZ decreased the expression of α5β1 integrin (as well as those of β3), while recovery of expression was found after removal of the drug. Interestingly, phenotypic alterations (proliferation and migration) are coupled with the level of this integrin. These findings portray α5β1 integrin as a promising target for recurrent GBM, as was proposed in studies examining bevacizumab-treated recurrent glioblastoma [37,60]. In our previous work [31], we showed that α5 integrin expression in primary tumours was not impacted by their MGMT status. Whether this absence of a relationship remains recurring is currently not known and deserves further studies.

Beside integrin changes, resistant cell lines exhibit p53 pathway activation with a long-lasting increase in MDM2 expression. Restoration of the tumour-suppressor function of p53 by disrupting the MDM2-p53 protein–protein interaction is considered an attractive therapeutic strategy for GBM expressing p53 wild type. Combination therapy of TMZ with Nutlin 3a, an MDM2 antagonist, was already shown to enhance the survival of mice engrafted with a GBM cell line by activating p53 and downregulating DNA repair proteins [61]. Preclinical evaluation of RG7112, another MDM2 antagonist, showed a reduced tumour growth of p53 wild-type patient-derived cell lines with the amplification of MDM2 [17]. The significant efficacy of this drug in a subset of non-MDM2-amplified models has also been observed. We show in this work that TMZ-resistant cells remain sensitive to MDM2 inhibitors and that the p53 pathway can be over-activated in these cells by three different drugs. Our previous results point out a negative crosstalk between α5β1 integrin and p53 signalling pathways implicated in chemotherapy resistance of glioma cells. Activating p53 concomitantly with inhibiting integrin α5β1 led to an increase in p53 signalling and glioma cell death in α5 integrin subunit-overexpressing cells [30,39]. We evaluated this strategy on the TMZ-resistant U87MG R50 OFF cells, which re-express the integrin. The results obtained in experiments associating Idasanutlin (the most efficient MDM2 antagonist) with FR248 (the most specific α5β1 integrin antagonist) suggest that resistance to TMZ may be overcome by this strategy. The U87MG R50 OFF cell line appeared more sensitive than the non-treated cells, even if the expression of the α5β1 integrin was similar between the two cell lines. We cannot exclude at this point that other molecular changes participate in this phenomenon, which deserves further studies. We thus propose a new therapeutic option for recurrent GBM expressing α5β1 integrin: p53 activation along with inhibition of the integrin. This therapy may be achieved with a single molecule, compound **9**, which is able to target RGD-integrin-expressing cells and inhibit MDM2 at the same time [42]. We show here that this molecule is able to target U87MG and TMZ-resistant U87MG R50 OFF cells at lower concentrations than integrin antagonists, suggesting decreased potential side effects. Future works have to be carried out to more precisely investigate this compound, which may be very interesting to treat recurrent glioblastoma.

In conclusion, our work shows a huge impact of Temozolomide on the integrin repertoire of U87MG cells. The integrin expressions appear highly switchable during the course of temozolomide treatment. Specific integrins may be particularly targetable at different time points of glioblastoma treatment and combination therapies evaluated according to their time-dependent expression. Although confirmation in patient-derived cell lines and other preclinical models is needed, our data add new evidence that α5β1 integrin has a role to play as a therapeutic target in recurrent glioblastoma.

## Figures and Tables

**Figure 1 cancers-14-00369-f001:**
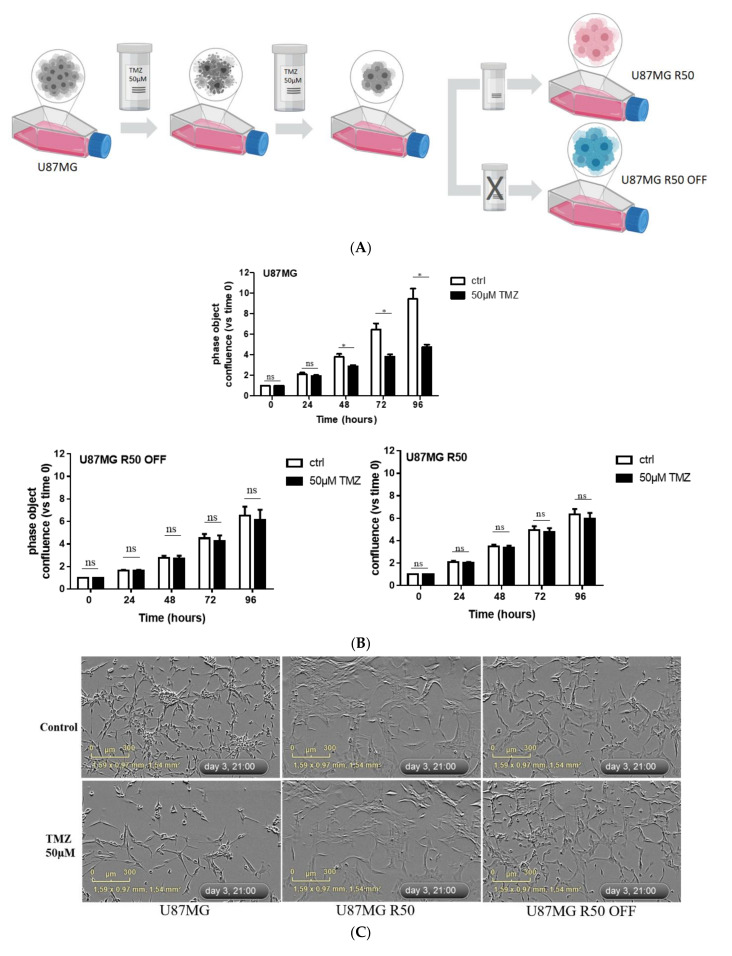
Confirmation of acquired TMZ resistance in U87MG GBM cells: (**A**) Generation of TMZ resistance U87MG R50 and U87MG R50 OFF cells. (**B**) Representative histograms showing the effect of TMZ (50 µM) on cell confluence of U87MG, U87MG R50 and U87MG R50 OFF cells from 0 to 96 h of treatment. Results are expressed as the relative area of plate covered versus the area covered by solvent-treated control cells. Histograms represent mean ± S.E.M. of at least three separate experiments. (**C**) Representative phase contrast images from the IncuCyte showing the cellular morphology of U87MG, U87MG R50 and U87MG R50 OFF cells 72 h post treatment with 50 µM of TMZ. Scale bar, 300 μm. *, *p* < 0.05; ns, non-significant.

**Figure 2 cancers-14-00369-f002:**
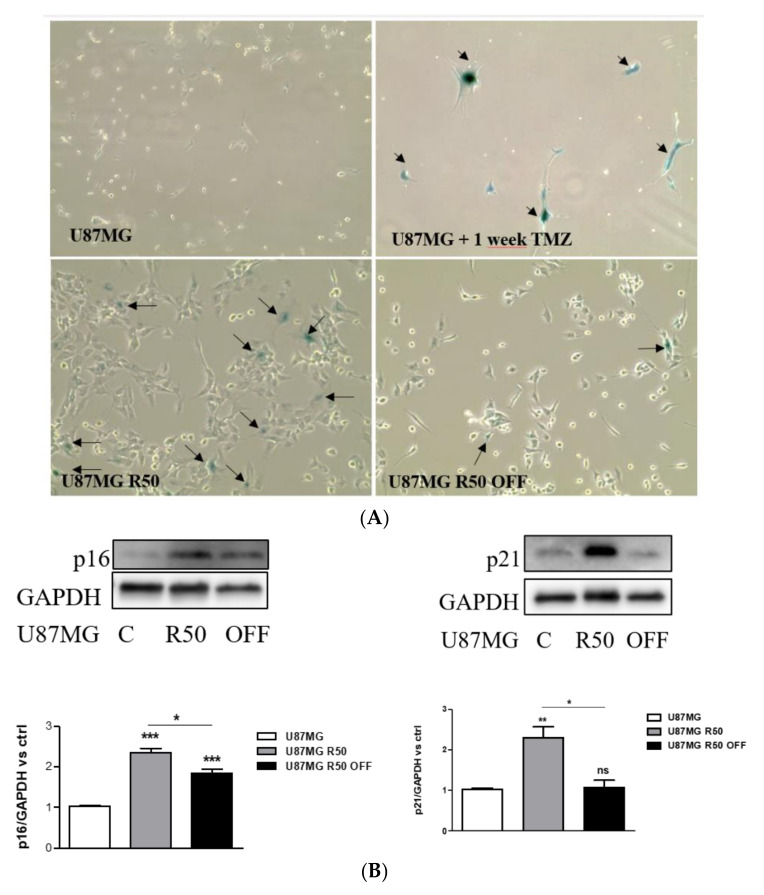
TMZ induces senescence in U87MG cells: (**A**) Representative photomicrographs showing cellular senescence after staining with SA-βGal. Arrows: senescent cells. (**B**). Representative Western blot analysis showing basal expression of p16 and p21 in TMZ-resistant cells (U87MG R50 and U87MG R50 OFF) compared to the parental cells. Histograms represent the mean ± S.E.M. of three separate experiments, and GAPDH expression was used as the loading control. *, *p* < 0.05; **, *p* < 0.01; ***, *p* < 0.001; ns, non-significant.

**Figure 3 cancers-14-00369-f003:**
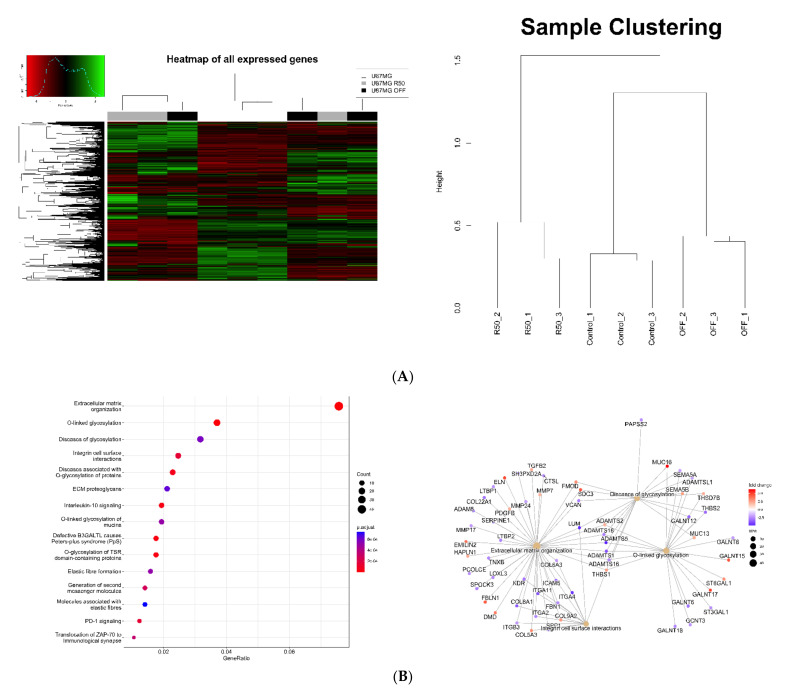
RNAseq analysis of TMZ-resistant and parental cell gene expression: (**A**) Heatmap grouped the nine samples based on the global expression profiles. Colour scale shows high and low expressions as green and red, respectively. Dendogram depicting correlation among different samples based on global expression profiles. (**B**) Top 15 enriched reactome pathways for differentially expressed genes in U87MG R50 cells versus control. Enrichment map with the inter-relation of the top three enriched reactome pathways and visualization of DEGs. (**C**) Top five enriched reactome pathways for differentially expressed genes in U87MG R50 OFF cells versus control. Enrichment map with the inter-relation of the top three enriched reactome pathways and visualization of DEGs.

**Figure 4 cancers-14-00369-f004:**
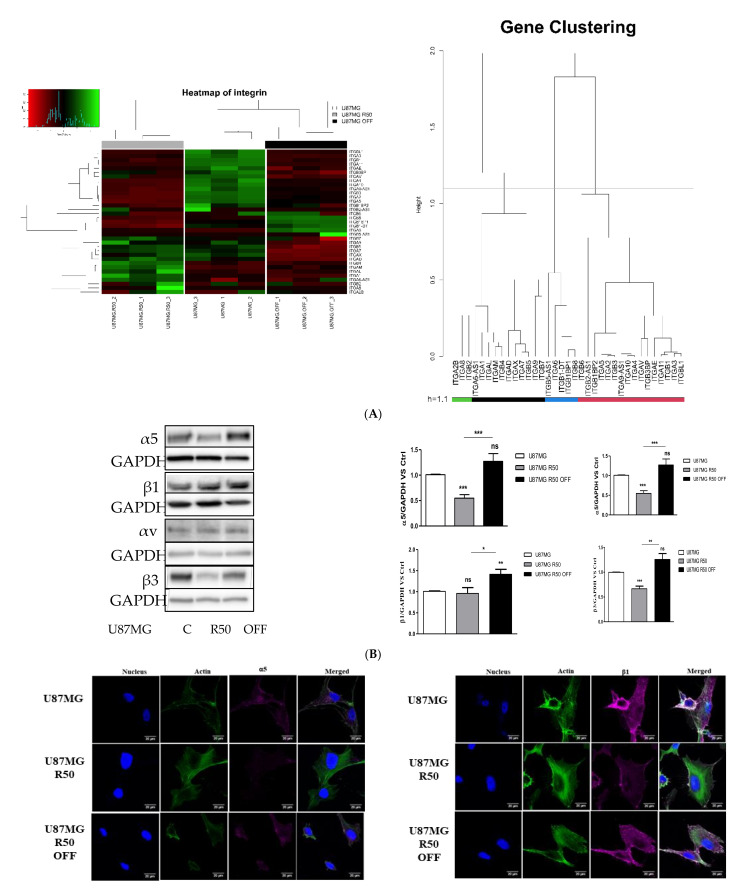
Integrin variations during TMZ treatment: (**A**) Heatmap visualization comparing expression of genes encoding integrins between samples with dendrogram to show clustering. Colour scale shows high and low expressions as green and red, respectively. (**B**) Representative Western blot analysis showing basal expression of α5, β1, αv and β3 integrins in U87MG R50 and U87MG R50 OFF cells compared to the parental cells. Histograms represent the mean ± S.E.M. of three separate experiments, and GAPDH was used as the loading control. (**C**) Representative fluorescence confocal microscopy images and mean grey value of basal α5 and β1 integrin subunits expression in TMZ-resistant U87MG R50 and U87MG R50 OFF cells compared to the parental cells. Scale bars: 20 µm. Histograms represent mean ± S.E.M. of three separate experiments. *, *p* < 0.05; **, *p* < 0.01; ***, *p* < 0.001; ns, non-significant.

**Figure 5 cancers-14-00369-f005:**
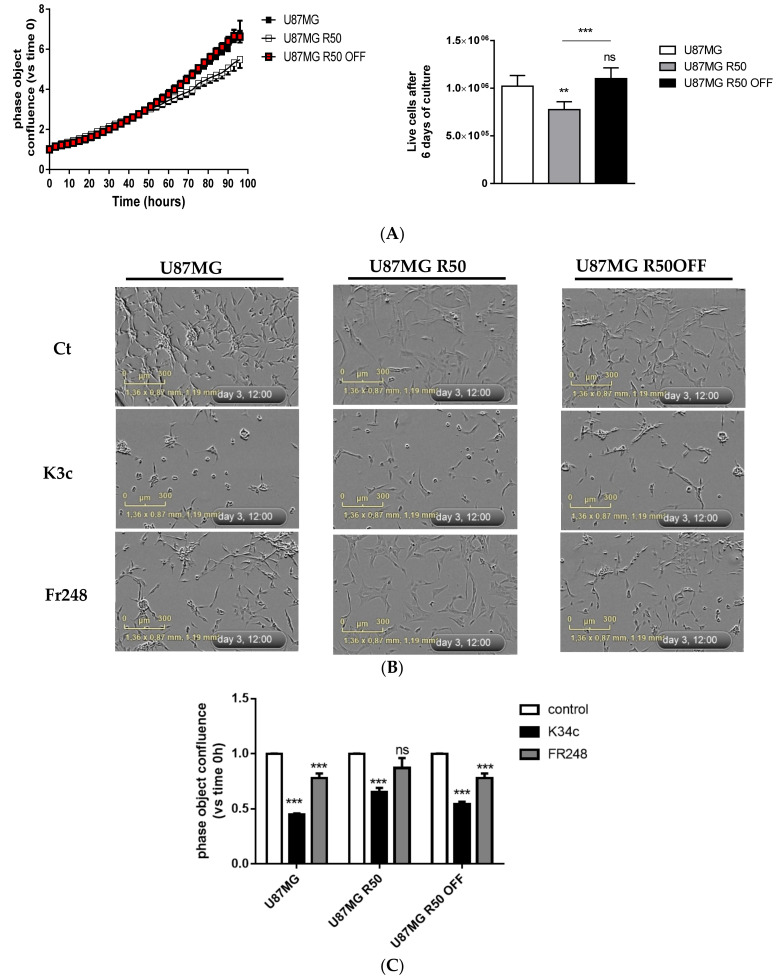
Proliferation and effects of integrin antagonists on TMZ-resistant and parental cells: (**A**) (left)**.** Cell confluence was calculated using IncuCyte Zoom software based on phase-contrast images of U87MG, U87MG R50 and U87MG R50 OFF cells from 0 h to 96 h. (right) Proliferation calculated as the number of viable cells after 6 days in culture compared to the number of plated cells at day 0. (**B**) Representative phase contrast images from the IncuCyte showing the cellular morphology of U87MG, U87MG R50 and U87MG R50 OFF cells 72 h post treatment with solvent, K34c (20 µM) and Fr248 (20 µM). Scale bar, 300 μM. (**C**) Cell confluence was calculated using IncuCyte Zoom software based on phase-contrast images of U87MG, U87MG R50 and U87MG R50 OFF cells at 72 h post treatment. For all panels: mean ± S.E.M. of at least three independent experiments with **, *p* < 0.01; ***, *p* < 0.001; ns, non-significant.

**Figure 6 cancers-14-00369-f006:**
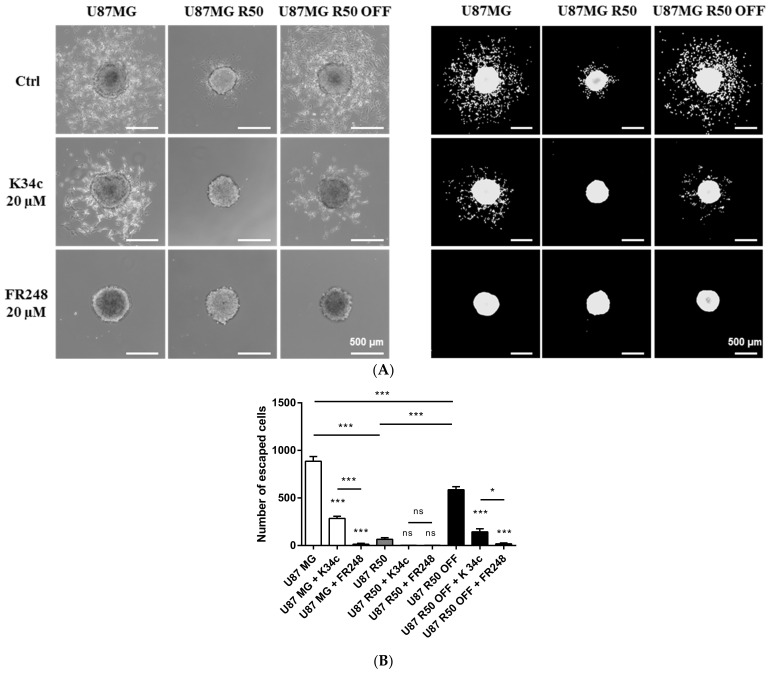
Cell evasion from spheroids and effects of integrin antagonists on TMZ-resistant and parental cells: (**A**) (left) Representative phase-contrast images of TMZ-resistant (U87MG R50 and U87MG R50 OFF) or parental cells spheroids after 18 h of migration and treatment with solvent, K34c 20 µM or Fr248 20 µM. Scale bars, 500 µm. (right) Representative fluorescence images (DAPI staining) of TMZ-resistant (U87MG R50 and U87MG R50 OFF) or parental cells spheroids after 18 h of migration and treatment with solvent, K34c 20 µM or Fr248 20 µM. Scale bars, 500 µm. (**B**) Analysis of the number of cells that migrated out of the spheroid and (**C**) analysis of the average distance of migration out of the spheroid for TMZ-resistant cells (U87MG R50 and U87MG R50 OFF) compared to parental cells, treated during 18 h with solvent, K34c 20 µM or Fr248 20 µM. Image analyses were performed with ImageJ software using a homemade plugin [43]. For all panels: mean ± S.E.M. of at least three independent experiments with *, *p* < 0.05; **, *p* < 0.01; ***, *p* < 0.001; ns, non-significant.

**Figure 7 cancers-14-00369-f007:**
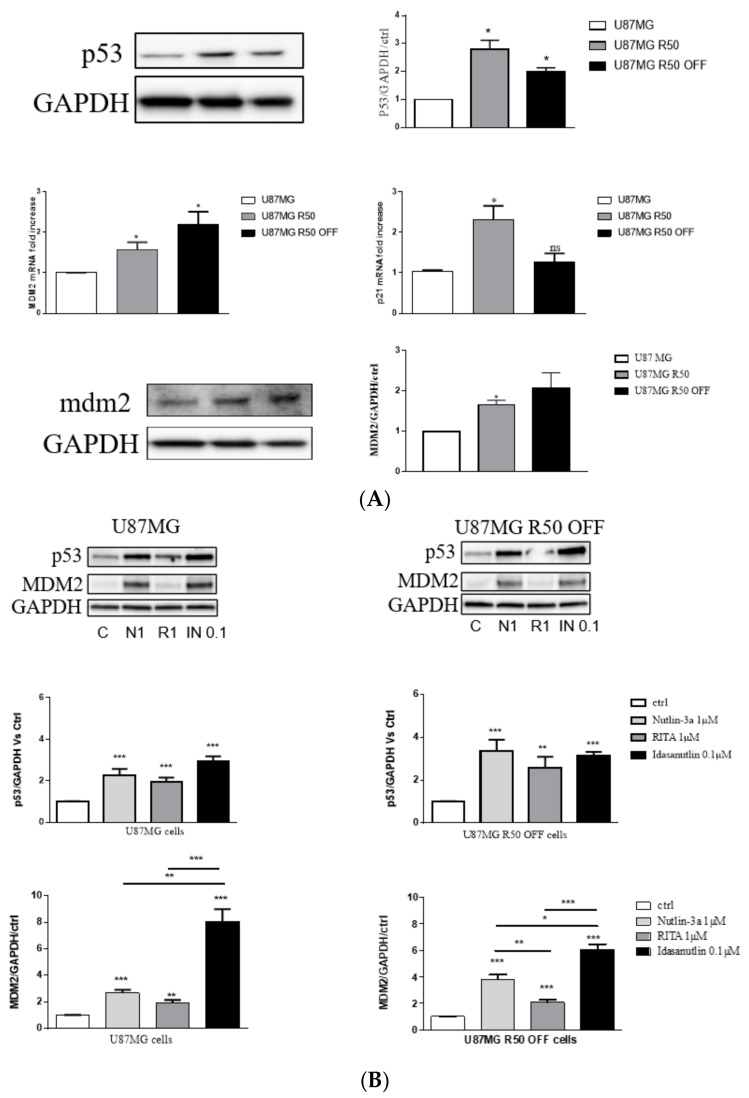
Modulation of the p53 pathway in U87MG, U87MG R50 and U87MG R50 OFF cells. (**A**) Representative Western blot of p53 (upper panel) and RT-qPCR analysis of p53 target genes mdm2 and p21 (middle panel) in U87MG R50 and U87MG R50 OFF compared to the parental cells. Representative Western blot of MDM2 is shown (lower panel) as well as the corresponding histograms. (**B**) Representative Western blot of p53 stabilisation and MDM2 expression 24 h post treatment with Nutlin-3a (1 µM/N1), RITA (1 µM/R1) and Idasanutlin (0.1 µM/IN 0.1) in U87MG R50 OFF cells compared to the parental cells. Histograms represent the mean ± S.E.M. of three separate experiments and GAPDH was used as the loading control. *, *p* < 0.05; **, *p* < 0.01; ***, *p* < 0.001; ns, non-significant.

**Figure 8 cancers-14-00369-f008:**
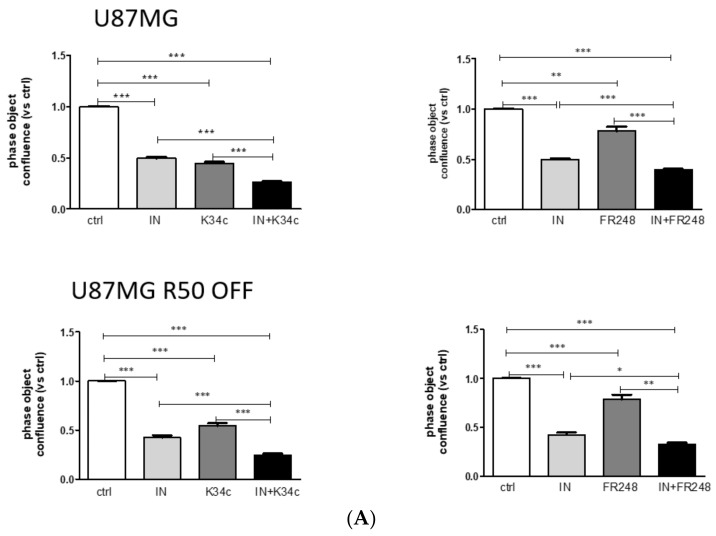
Effect of Idasanutlin and integrin antagonists on TMZ-resistant and parental cells. (**A**) Cell confluence was calculated using IncuCyte Zoom software based on phase-contrast images of U87MG and U87MG R50 OFF cells at 72 h after treatment with solvent, Idasanutlin (0.1 µM), K34c (20 µM) or FR248 (20 µM) alone or in combination. (**B**) Representative Western blots of p53 stability and activity (phosphorylation at ser^15^) and the p53 target gene MDM2 expression in U87MG R50 OFF compared to the parental cells 24 h post treatment with Idasanutlin (0.1 µM) and Fr248 (20 µM) either separately or in combination. Histograms represent the mean ± S.E.M. of three separate experiments with GAPDH as the loading control. For all panels: *, *p* < 0.05; **, *p* < 0.01; ***, *p* < 0.001; ns, non-significant.

**Figure 9 cancers-14-00369-f009:**
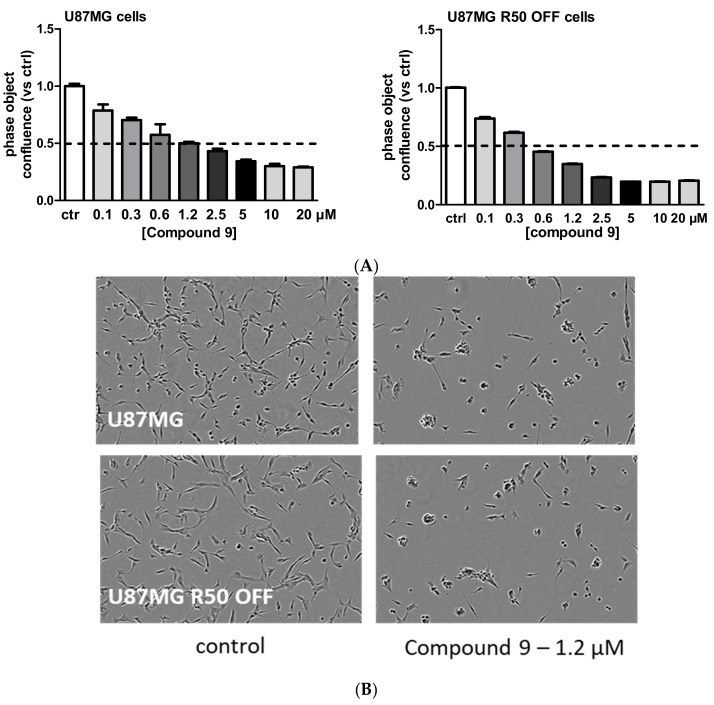
Effects of compound **9** on TMZ-resistant and parental cells: (**A**) Dose—response effects of compound **9** on cell confluence in U87MG and U87MG R50 OFF cells. (**B**) Representative phase contrast images from the IncuCyte showing the cellular morphology of U87MG and U87MG R50 OFF cells 72 h post treatment with compound 9 (1.2 µM). Histograms represents the mean ± S.E.M. of three separate experiments after 72 h treatment with TMZ or compound **9**.

**Table 1 cancers-14-00369-t001:** List of antibodies used in the study.

Antibody	Blocking Solution	Antibody Dilution
Anti-α5 integrin H104 (Santa Cruz)	5% milk/1 × TBS/0.1% Tween -20	1/1000
Anti-β1 integrin TS2/16 (Millipore)	5% milk/1 × TBS/0.1% Tween -20	1/1000
Anti-αv integrin (Cell signalling)	5% milk/1 × TBS/0.1% Tween -20	1/1000
Anti-β3 integrin (Cell signalling)	5% milk/1 × TBS/0.1% Tween -20	1/1000
Anti-p53 (BD Bioscience)	5% milk/1 × TBS/0.1% Tween -20	1/1000
Anti-pp53ser15 (Cell signalling)	5% BSA/1 × TBS/0.1% Tween -20	1/1000
Anti-p16 (Cell signalling)	5% milk/1 × TBS/0.1% Tween -20	1/1000
Anti-p21 (Cell signalling)	5% milk/1 × TBS/0.1% Tween -20	1/1000
Anti-MDM2 (Calbiochem)	5% milk/1 × PBS/0.1% Tween -20	1/1000
Anti-GAPDH (Millipore)	5% milk/1 × TBS/0.1% Tween -20	1/5000
Anti-tubulin (Sigma-Aldrich)	5% milk/1 × TBS/0.1% Tween -20	1/3000
Mouse HRP-conjugated secondary antibody	5% milk/1 × TBS/0.1% Tween -20	1/10,000
Rabbit HRP-conjugated secondary antibody	5% milk/1 × TBS/0.1% Tween -20	1/10,000

**Table 2 cancers-14-00369-t002:** Summary of Incucyte experiment results at 3 days after treatment with the different drugs alone or in combination. Results are expressed as mean ± s.e.m. of 3 to 5 independent experiments. * refers to statistical comparison between control and treatment and # comparison between combination therapies and each treatment alone. * and # are indicative of *p* < 0.01. ns = non significant.

Treatment	U87MG Cells	U87MG R50 OFF Cells	U87MG R50
Control	1	1	1
TMZ	0.57 ± 0.01 *	0.94 ± 0.02 ns	-
K34c	0.44 ± 0.01 *	0.54 ± 0.03 *	0.65 ± 0.04 *
FR248	0.78 ± 0.04 *	0.78 ± 0.04 *	0.86 ± 0.09 ns
Nutlin-3a	0.35 ± 0.01 *	0.42 ± 0.02 *	0.42 ± 0.01 *
Idasanutlin	0.52 ± 0.05 *	0.45 ± 0.02 *	_
Nutlin-3a + K34c	0.21 ± 0.01 * #	0.24 ± 0.01 * #	0.25 ± 0.008 * #
Nutlin-3a + FR248	0.28 ± 0.02 * #	0.33 ± 0.01 * #	0.37 ± 0.01 * ns
Idasanutlin + K34c	0.265 ± 0.005 * #	0.244 ± 0.007 * #	-
Idasanutlin + FR248	0.399 ± 0.005 * #	0.319 ± 0.009 * #	-

## Data Availability

RNAseq data will be available after request to the corresponding author.

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
