# Peer review of "Temozolomide-Acquired Resistance Is Associated with Modulation of the Integrin Repertoire in Glioblastoma, Impact of α5β1 Integrin"

_cancers, 2022, doi:10.3390/cancers14020369_

Round 1

Reviewer 1 Report

Sani et al. investigated changes in integrin expression related to TMZ exposure in parental and TMZ-resistant U87MG cell lines and how these changes might be therapeutically targeted. The results of the study are sound and clearly presented and might have future therapeutic implications. I compliment the authors for their fine work. I have just few minor questions.

- Was MGMT methylation status assessed in U87MG cells and, more importantly, in U87MG R50 and U87MG50 OFF cells? There are some data in literature suggesting that recurrent tumors might change their MGMT promoter methylation status.

- Do the authors expect their results to apply, to some extent, also to MGMT unmethylated GBM? It is not uncommon in clinical practice to offer TMZ to MGMT unmethylated patients if they're young and have good performance status.

Author Response

We thank the reviewer for this important remark. Indeed, the status of MGMT is of importance in recurrent tumors. We did not check the methylation status of MGMT but according to the RNAseq data we do not see an increase in mRNA level of MGMT in the two resistant cell lines suggesting that a demethylated MGMT promoter is not involved in these cell resistance to TMZ. We added this information in the result part (lane 278/279, p6).

We can expect that TMZ-induced integrin expression variations but also answer to p53 reactivation strategy may be applied to MGMT unmethylated tumors as no clear link can be drawn at the time between integrins and MGMT status. In our previous work on clinical samples (Etienne-Selloum et al, 2021, Pharmaceuticals; ref 32), we did not observe significant differences between methylated and unmethylated MGMT primary tumors concerning the α5 integrin expression. It would be interesting to check this point in recurrent tumors. We added this point in the Discussion part (lane 609-611, p19).

Reviewer 2 Report

The author established TMZ-resistant cells by subjecting U87MG cells to TMZ 50μM treatment as an in vitro model of MGMT-negative TMZ resistance and evaluate the molecular and phenotypic changes occurring during and after TMZ treatment in this resistant model. The author found that the dynamic expression changes of α5β1 integrin and P53 may be important to TMZ resistance and propose an alternative to alter the viability of recurrent glioblastoma cells expressing high level of α5β1integrin. Here are my listed comments.

Major points:

Point 1: Figure 1A: How long did the author maintain the U87MG R50 OFF without TMZ before measuring the effect of TMZ on cell confluence. If the maintaining time is very short, it is not surprising that U87MG R50 OFF still show resistance to TMZ.

Point 2: Figure 1B: how to explain the U87MG R50 OFF with a mixed morphology in Figure 1B, but still show the same resistance result as U87MG R50 in Figure 1A. The author should address culture condition clearly, as all the results in the manuscript are based on the maintaining condition.

Point 3: Figure 7A: The Wb result of p21 and MDM2.

Point 4: Figure 7B: How about the result in U87 MG R50 cells?

Point 5: Figure 8: According to Figure 7, For the activator of P53, RITA is better than Nutlin3a, and Both of Nutlin3a and Idasanutlin bind to MDM2 part. Why the author not choose RITA? Is there any explanation?

Point 6: Figure 8: The figure legend is “Effect of Idasanutlin and integrin antagonists on TMZ-resistant and parental cells. Why there is no data to show the result on U87 MG R50 cells? Please add it.

Point 7: Figure 9: There is no data to show the result on U87 MG R50 cells

Point 8: Is there any regulation between α5β1 and P53? This can be discussed in the discussion.

Minor points:

Point 1: The A, B, C, D... in each Figure are placed in different position, which decrease the readable of the figure. What is more, some labeled with bold, some are not as well as some with parentheses around, some are not. Please adjust them to be consistent. Some panels include several small panels, would it be better to add more label to descripe.

Point 2: Is there any difference between U87 MG R50 OFF and U87 MG OFF R50 throughout the manuscript?

Point 3: Figure 2B: The font size of the label in y-axis is not consistent.

Point 4: Figure 3A: The font size in right panel is small to read.

Point 5: Figure 3B: The font size in right panel is small to read.

Point 6: Figure 3C: The font size in right panel is small to read

Point 7: Figure 4A: The font size in right panel is small to read.

Point8: Figure 7B: The x-axes are not aligned in MDM2/GAPDH/ctrl.

Author Response

We thank the reviewer for all the remarks.

Point 1 and Point 2.

The U87MG R50 OFF cells are maintained in TMZ-free medium since several months. We check regularly (about every 3 months) by the Incucyte technology their resistance status which remains stable in our hands.  This point was added in the Material and Methods part (lanes 152-154, p4).  Very few U87MG R50 OFF cells appear morphologically senescent as shown in the Figure 2 and we believe that this low percent do not allow to see any difference versus control cells in proliferation assays.

Point 3. Figure 7A.

The WB results for p21 are shown in the Figure 2. Text was added to refer to this figure (lane 454/455, p14).

The WB for mdm2 was added in Figure 7A. Text was added to refer to this figure (lane 454/455, p14) and legend to the figure was modified.

We did focus on the cell lines expressing α5 integrin as combination therapies or effect of compound 9 included specific inhibition of this integrin. We apologize for the absence of results on U87MG R50 cells and ask the referee not to include them as the delay to answer is too short to make correctly these experiments.

Point 5, Figure 8.

According to Figure 7, RITA was the less efficient drug for p53 activation in the 3 cell lines. We therefore focused mainly on Nutlin and Idasanutlin which proved able to stabilize and activate p53 even more rapidly than RITA (kinetic data not shown). In addition, Idasanutlin appeared interesting as it works at a concentration 10 times lower (0.1 µM) than those of Nutlin and RITA (1 µM).

Point 4. Figure 7B, Point 6, Figure 8 and Point 7, Figure 9

We did focus on the cell lines expressing α5 integrin as combination therapies or effect of compound 9 included specific inhibition of this integrin. We apologize for the absence of results on U87MG R50 cells and ask the referee not to include them as the delay to answer is too short to make correctly these experiments. We nevertheless included in the first version of the manuscript the results of compound 9 on U87MG- α5-KO cells (Supplementary figure 5) to show that the absence of this integrin expression led to a shift in the EC50 of the drug (0.6 µM for U87MG and U87MG R50 OFF cells versus 2.5 µM for U87MG-α5Ko). Compound 9 clearly affects α5-expressing cells more efficiently than non-expressing cells.

Point 8.

In previous works (Janouskova et al, Cancer Research, 2012 and Renner et al, Cell Death and Diff, 2015) we analyzed the impact of α5β1 integrin on p53 pathway and inversely using U87MG cells over-expressing (by genetic manipulations) the α5β1 integrin. We showed the existence of a negative cross regulation between both pathways and that inhibiting the integrin and activating p53 by Nutlin was an effective combination to kill the cells. In the present work, we enlarged this concept to TMZ-resistant U87MG R50 OFF cells which we propose as a model of some kind of recurrent tumors. Even if α5β1 integrin is not overexpressed (as compared to control cells) in these resistant cells, inhibition of the integrin seems to sensitize them to Idasanutlin-dependent p53 activation (Figure 8).  Presumably other molecular changes beside the inhibition of the re-expressed integrin may be involved in this sensitization. We added a sentence about this point in the discussion part (p19, lanes 630/633).

Minor points

Point 1

We apologize for the bad presentation of figures in the manuscript. We made corrections for this point in the revised version to make the figures clearer.

Point 2

We apologize for this mistake. We homogenized in the revised version the name to U87MG R50 OFF cells.

Points 3 to 8

We made all the changes asked by the reviewer.

Reviewer 3 Report

All data are very low quality. In this quality, I can't understand results which the authors explain. In addition, I don't feel important differences between parental U87MG cells and U87MG R50/R50 OFF cells. My understanding is only to increase  integrin alpha5 expression in U87MG R50 OFF. What is the important effects in glioma cells after TMG treatment?

Attached PDF files show clear figures, but I don't change my reject decision. Because I can't find the novelty and author's main claim in this study. There is not important information for glioma cells after TMZ drug treatment, which have the same effects with non-treated glioma cells.

Author Response

Reviewer 3

We apologize for the low quality of figure presentations in the first version. We corrected this point in the revised version.

We believe that different points may be of interest for GBM knowledge.

In our knowledge, the huge variations in integrin expressions during and after TMZ treatments was never described. The α5β1 integrin was chosen as an example as we demonstrated (as well as others) that it is implicated in the worse outcome of patients (ref 32, Etienne-Selloum et al., Pharmaceuticals, 2021). Other integrins may be as well important as they also showed variations of their expression. One of the point that we wanted to underline is that treatment with integrin antagonists may be adjusted to tumors expressing the integrins of interest. The clinical trials with cilengitide confirmed this point as the post evaluation of CENTRIC and CORE trials showed a better answer in CORE patients expressing αvβ3 integrin in their tumoral cells at diagnosis (Weller et al, Oncotarget, 2016).  We show here that TMZ is able to decrease several integrins including α5β1 integrin but that removal of the drug allows re-expression of this integrin. This suggests that integrin inhibition concomitantly with TMZ will be less effective than after TMZ washout. We addressed this point in the discussion part “The heatmap of integrins (Figure 4) revealed particular sets of integrins overexpressed in each cell line suggesting that anti-integrin therapeutic options have to be considered in a timely manner during therapies.” We agree that re-expression of the integrins may be confirmed in other preclinical models as well as in clinical samples of recurrent tumors. We already showed that expression of α5 integrin in primary tumor samples is highly heterogeneous between patients and that a high expression alters the patient survival after completion of the Stupp protocol (Etienne-Selloum et al, Pharmaceuticals, 2021). The link between a high expression in the primary tumor and at recurrence remains to be demonstrated.

The reviewer is right to say that there are no gross differences between control cells and resistant R50 OFF cells as shown in the sample clustering (Figure 3A). But one major difference is the sensitivity to TMZ. We propose an imperfect model (based on U87MG cells) of tumors recurring after treatment, being insensitive to the first line treatment but able to answer to another treatment. We already proposed the association of integrin antagonist and p53 activators as new option for GBM (ref 31, Janouskova et al, Cancer Res, 2012 and ref 40 Renner et al, Cell death and Diff, 2015) highly expressing the α5β1 integrin.  In this paper, we propose to use this combination in TMZ resistant cells which in our knowledge has never been proposed. We are aware that such therapy will not be effective for all GBM according to their high molecular heterogeneity at diagnosis and after recurrence. The expression of α5β1 integrin itself is highly heterogeneous between tumors and locally in a given tumor. The drivers of this expression are currently under investigations in our laboratory. Our therapy proposal will concern only primary tumors expressing the integrin with a WT p53 status at diagnosis but also some TMZ-resistant presumably recurring tumors.

Reviewer 4 Report

In this study, the authors investigated modification of glioblastoma cancer cells which became resistant to temzolomide.  They show that some integrins are decreased during temozolomide treatment, and restored when the drug is suspended, proposing that they could represent potential therapeutic targets in glioblastomas resistant to temzolomide.

They also show alterations in TP53/MDM2 pathway, consisting in MDM2 increase, proposing MDM2 inactivation as a further potential therapeutic strategy in glioblastoma resistant to temzolomide.

The study is interesting and well designed and provides original findings. However, the results on "p53 signalling pathway in U87MG and TMZ-resistant cells" are unclear. Is there a difference in P53 expression between the three cell lines? I suggest to re-write this paragraph.  

Author Response

Reviewer 4

We thank the reviewer for the remark about p53 signaling pathway. In fact, as indicated in the Figure 7A, U87MG R50 and U87MG R50 OFF cells express a stabilized p53 and an increase in p53 target genes (such as mdm2) at basal level. Level of mdm2 (at the protein level) was rather increased by Idasanutlin in U87MG control and R50 OFF cells (Figure 8B) confirming the reactivation of p53 signaling. In previous works (Janouskova et al, Cancer Research, 2012 and Renner et al, Cell Death and Diff, 2015) we analyzed the impact of α5β1 integrin on p53 pathway and inversely the impact of p53 on α5β1 integrin, using U87MG cells over-expressing (by genetic manipulations) the α5β1 integrin. We showed the existence of a negative cross regulation between both pathways and that inhibiting the integrin and activating p53 by Nutlin was an effective combination to kill the cells. In the present work, we enlarged this concept to TMZ-resistant U87MG R50 OFF cells which we propose as a model of some kind of recurrent tumors. Even if α5β1 integrin is not overexpressed (as compared to control cells) in these resistant cells, inhibition of the integrin seems to sensitize them to Idasanutlin-dependent p53 activation (Figure 8). Presumably other molecular changes beside the inhibition of the re-expressed integrin may be involved in this sensitization. We added a sentence about this point in the discussion part (p19, lanes 620-623).

To be more clear we split this paragraph in two parts one showing the p53 signaling at basal level and after activation with Nutlin, RITA and idasanutlin (part 3.5) and one showing the combination therapy with p53 activation and integrin inhibition (part 3.6).

Round 2

Reviewer 2 Report

The author answered my points.

Author Response

Thank you very much for your comments.

Reviewer 3 Report

I understood the author's claim. I reviewed revised version again.

comments

Figure 2A. Could you edit the brightness of U87MG OFF R50’s image as well as other three panels?

Figure 4B. Under the Western Blotting data, “U87MG C R50 OFF” is indication error.

Figure 7B right pannel. In R1-treat sample lane, the author should re-try SDS-PAGE&blotting about only anti-p53. Because I judged that only this band is incorrect blotting data with the air.

line 29. “in reccurent tumors.__Specific-“ There may be double space.

line 65-66. What is “cellular accumulation”? Does this mean “intracellular accumulation”?

line 165,173,225,226,228,229. The author should insert a space. “200_ul”, “24_hours”etc…

line 220. “25 000 cells” should be changed to “25,000 cells”.

Author Response

We thank the reviewer for its positive comments.

Reviewer 4 Report

I have no other comments

Author Response

Thank you very much for your comments.